# METATOOL: FACILITATING LARGE LANGUAGE MODELS TO MASTER TOOLS WITH META-TASK AUGMENTATION

## ABSTRACT

Utilizing tools with Large Language Models (LLMs) is essential for grounding AI agents in real-world applications. The prevailing approach involves few-shot prompting with demonstrations or fine-tuning with expert annotations. However, mere in-context demonstrations may fail to cover sufficient knowledge for complex tools and tasks. Training on solution paths is also hindered by the high cost of expert annotations and generalizing to new tools. A core challenge of generalizable tool use lies in understanding the "meta", or fundamental natures of tools that are transferable across tasks, such as causality and constraints. In this paper, we present *MetaTool*, a novel tool learning methodology designed to generalize across any reusable toolset. Our approach incorporates a self-supervised augmentation technique derived from a series of meta-tasks. This involves predicting masked elements in the tool execution process. The self-supervised procedure enables scalable generation of high-quality QA data, which is handy for supervising tool understanding. By incorporating meta-task data into task-oriented training, our method significantly enhances the performance of open-source LLMs, achieving results comparable to ChatGPT in both tool-based planning and chatting scenarios. Through large-scale instruction tuning, the MetaTool model demonstrates impressive zero-shot generalizability on new tasks.

## 1 INTRODUCTION

Distinguished from other species, a unique characteristic of human advanced intelligence is using complex tools, which expands the frontiers neural intelligence can reach. With the advent of powerful foundation models, AI has the potential to solve complex tasks with these external mechanisms. LLMs have been majorly oriented towards either tool-augmented chatbots equipped with retrievers and search engines, or tool-oriented agents (e.g. web navigation Rawles et al. (2023); Hong et al. (2024), embodied manipulation Chi et al. (2023)) that achieve task objectives through tool output Qin et al. (2023b). While the former emphasizes generalizing to various tools, the latter focuses on complex tools and scenarios.

To efficiently integrate LLMs with tools, a mainstream way relies on in-context learning (ICL). The model is provided with the "cookbook" of tools in zero-shot prompting or demonstrations in few-shot prompting Xu et al. (2023); Paranjape et al. (2023); Brown et al. (2020). It may work well on simple tools with frameworks like LangChain Chase (2022). However, for complex tasks with sophisticated tools, in-context learning is limited that demonstrations can not exhaust all scenarios, and manuals are also limited in length. Ultimately, it's impractical to expect LLMs to be intelligent enough to master any tool without the experience of using it. On the other side, training-based methods Qin et al. (2023c); Patil et al. (2023); Dubey et al. (2024) mainly adopt supervised fine-tuning with annotated expert solutions. Despite the difficulties in scaling up the optimal annotation, supervision with task solutions has limitations. Task-agnostic knowledge of tools can be neglected, which hinders the generalization to diverse scenarios or new tools. Self-play training methods like Toolformer Schick et al. (2024) and TALM Parisi et al. (2022) integrate the inference process with

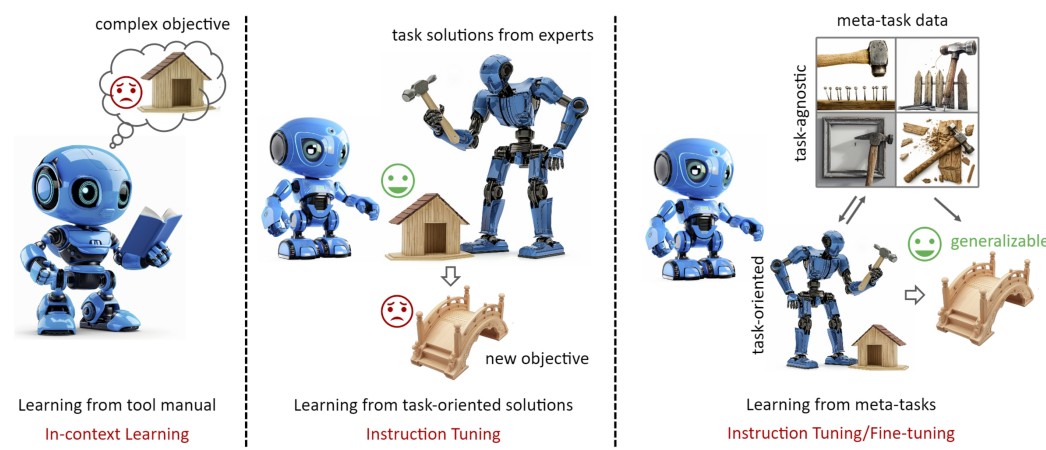

Figure 1: **Paradigm comparison** between existing tool learning methods and proposed meta-task augmentation. While the prevailing methods are limited in generalizing to complex scenarios or new tools, MetaTool enables gaining transferable tool understanding from task-agnostic knowledge.

self-supervised tool calling data. Although such a manner maintains the generality of tool calling, it's constrained in question-answering scenarios.

Empirically, human learners get familiar with tools such as hammers (e.g., for nailing and smashing) before engaging in complex construction. Generalizable tool use should be achieved based on the fundamental understanding of tools themselves that holds stable for different objectives, namely task-agnostic (illustrated in Figure 1. Naturally the formation of tool understanding can be disentangled from the learning of task solving. In this paper, we introduce *MetaTool*, a general methodology that enables both complex tool mastery and unseen tool generalization on top of task-agnostic tool understanding. We design a set of meta-tasks inquiring about the causality of the toolset as an autonomous system and its functionality as a function. Given a callable toolset (e.g. APIs, functions), meta-task data is constructed in a scalable self-supervised way based on unsupervised or self-play tool executions. Augmenting task-oriented training with meta-task data, LLMs learn to solve problems while deepening the mastery of tools. We conduct experiments on both complex tool-oriented tasks and tool-augmented benchmarks, demonstrating that MetaTool significantly exceeds models trained merely on annotated solutions in both worlds and is competitive with the latest LLMs (ChatGPT) with the size of 8B. The overall contribution can be summarized in three-folds:

- We introduce a new tool learning method that facilitates LLMs to master tools with task-agnostic tool understanding.

- We propose an integral set of self-supervised meta-tasks that dissect the tool execution process. Meta-tasks enable expert-free data generation and augmentation across tool-augmented and tool-oriented scenarios.

- Extensive evaluation on both tool-oriented tasks and tool-augmented benchmarks demonstrates the effectiveness and generality of MetaTool, narrowing the gap between open-source models and state-of-the-art LLMs.

## 2 APPROACH

In this section, we first formalize the task of using a close toolset and define 6 general meta-tasks that are key to tool understanding. Then we show how to construct datasets in an integral self-supervised way covering different scenarios. In the end, we describe several schemes to augment tool learning with meta-tasks.

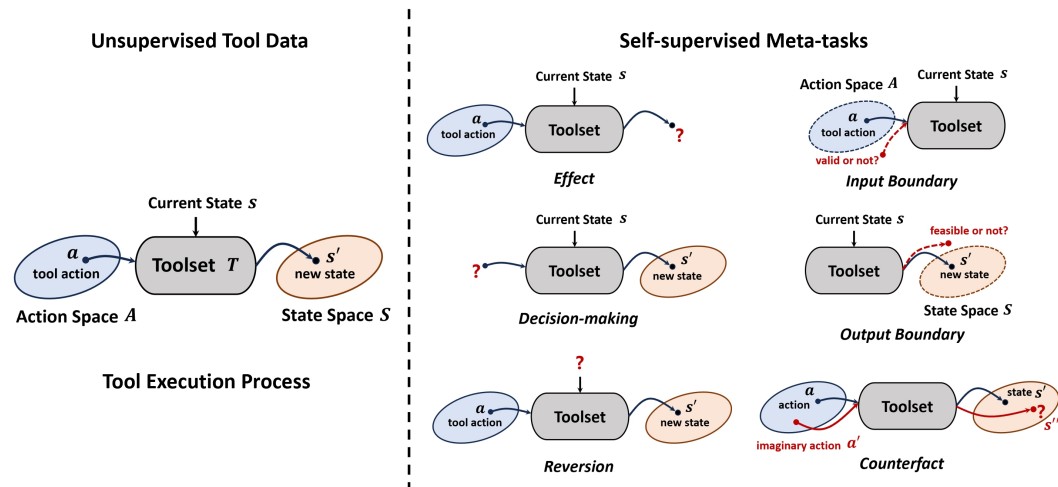

Figure 2: Illustration of developing self-supervised meta-tasks from unsupervised tool execution process.

## 2.1 SELF-SUPERVISED META-TASKS FOR TOOL UNDERSTANDING

**Problem formalization.** A tool-use task can be generally defined as a Markovian tuple $\langle \mathcal{S}, \mathcal{A}, \mathcal{T}, g \rangle$, where $\mathcal{S}, \mathcal{A}, \mathcal{T}$ is the state space, action space, and toolset, and $g$ is the goal state of the task. Toolset $\mathcal{T} = \{t\}_N$ consists of $N$ tools, each as a state transition function $s' = t(s, \theta)$ that formalizes the outcome of state change when feeding the input parameters $\theta$ into the tool. An action $a = \langle t, \theta \rangle \in \mathcal{A}$ specifies the tool and its input. As an autonomous agent, an LLM should iteratively respond with tool calling and inputs according to the state until it reaches the goal. A solution path leading to the goal can be defined as a sequence of actions and states $p = \{s_1, a_1, ..., s_T, a_T\} \in P$. Broadly, when the tools can not alter any external state, tool output like retrieval results can be regarded as the state, and the desired information is the goal $g$.

We enhance the tool understanding of the model with self-supervised surrogate (pretext) tasks instead of in-context descriptions or demonstrations. Formally, we regard tools as external systems that implement state transition mappings. Tool understanding, therefore, involves comprehending the perception-action process of these systems (referred to as tool execution) and should be generalizable to various task objectives.

**Meta-task definition.** We begin with single-step tool execution $\mathcal{D} = \{s, a, s'\}$, peeling off the task goal $g$ context, which results as unsupervised data. Six surrogate tasks (meta-tasks) are designed based on the dataset $\mathcal{D}$. Basically, the model is required to predict masked elements of the execution process. It's similar with the idea of masked language models Devlin (2018) in a broader and structural granularity to learn the lurking knowledge beneath the unsupervised data. We define the meta-tasks as below (Figure 2):

- **Effect**: The model predicts the outcome state $\mathrm{P}(s'|a, s)$ given the initial state and the action.
- **Decision-making**: The model decides a feasible action $\mathrm{P}(a|s, s')$ given the initial and outcome state.
- **Reversion**: The model deduces the initial state $\mathrm{P}(s|a, s')$ given the action and the outcome state.
- **Input Boundary**: The model determines whether an action can be successfully executed, namely whether the action falls in the valid action space, given the current state: $\mathrm{P}(\mathbb{1}_{s' \neq s}|a, s)$.
- **Output Boundary**: The model determines whether a state can be reached with any action given the current state: $\mathrm{P}(\mathbb{1}_{\exists (t, \theta), s' = t(s, \theta)}|s, s')$.
- **Counterfact**: The model predicts the new outcome state $\mathrm{P}(s''|a, s', a')$ if a new action $a'$ were executed given that the current action $a$ results in the current outcome $s'$.

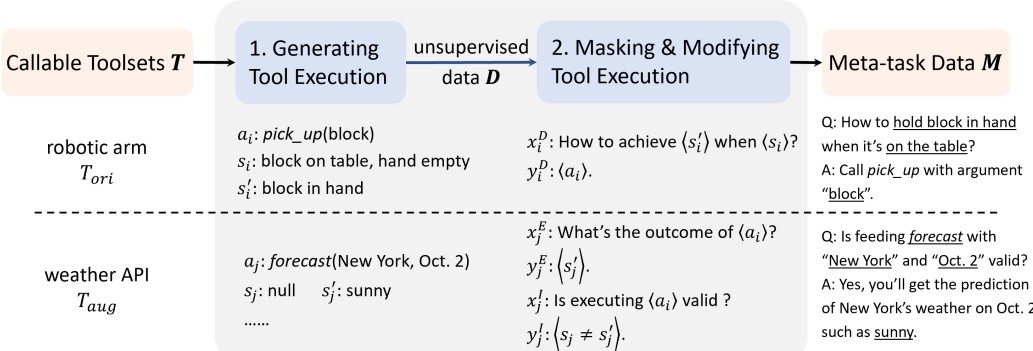

Figure 3: **Two-step approach to construct metaset.** We illustrate two exemplified processes of both tool-oriented and tool-augmented scenarios, which don't require any expert annotation. $x_i^D, y_i^D$ denotes the $i$-th question-answer pair of *decision-making* meta-task, et cetera.

*Effect*, *decision-making*, *reversion* meta-tasks emphasize the causality of a tool, regarding the action as the intervention to the state Pearl (2009); Pearl & Mackenzie (2018) and the outcome as the causal effect is determined by the tool mechanism. On top of that, *counterfact* task is the composition of *reversion* and *effect*, further imagining the outcome altered from the fact in *effect* task. This meta-task raises higher requirements on counterfactual reasoning Bareinboim et al. (2015); Zhang & Bareinboim (2016), an advanced form of causal reasoning that humans use to contemplate 'what if'. Moreover, tools implemented as APIs may receive non-executable inputs and result in ineffective outcomes. Thus the input and output domains are also unique features of a tool as a function. We consider *input boundary* meta-task that emphasizes the tool affordance that refers to what actions can be executed considering the situation and the precondition. *output boundary* meta-task emphasizes the functionality of tools, that is, what goals can and cannot be achieved given the current state.

## 2.2 METASET CONSTRUCTION

Based on the definitions, the dataset of meta-tasks (referred to as metaset) is generated as question-answering pairs to maintain the conversational skills of LLMs. To answer the meta-task questions, the trained model needs to understand the toolset mechanisms from the corresponding aspects. Given a set of reusable and callable tools $T$, the metaset $\mathcal{M} = \{x_n^m, y_n^m\}_{n=1}^N$ can be constructed in two steps, as illustrated in Figure 3, where $x_n^m, y_n^m$ is the $n$-th question-answer pair of meta-task $m$. First, we generate tool execution data $\mathcal{D}$ with the toolset. For a limited amount of tools and state space, stochastic sampling can be applied to initialize state $s \sim \mathrm{P}(\mathcal{S})$ and action $a \sim \mathrm{P}(\mathcal{A})$, and obtain the tool output $s'$. For large toolsets and diverse task scenarios that are hard to enumerate, we incorporate LLMs with self-play or tree search techniques to reduce redundant trials $a, s' \sim \mathrm{P}(\mathrm{LLM}(s, g))$. We prompt the LLM to also generate a "thought" analyzing the situation and what to do for each action following ReAct Yao et al. (2022), to elicit the reasoning ability. Thus an action includes the tuple of thought-tool-input. Note that the LLM does not need to be proficient in tool tasks, as the execution data $\mathcal{D}$ is irrelevant to the task performance. Non-executable actions also contain valuable knowledge and can be transferred as invalid samples in *input boundary* meta-task. The tool execution data should be sufficient to cover the various scenarios that may arise during the tasks.

Second, for the $n$-th sample and meta-task $m$, we insert the variables of states and actions into $K$ sets of templates (diversified with GPT-4) to obtain diverse QA data.

$$(x_n^m, y_n^m) = \mathrm{Mask}(a_n, s_n, s_n', \Phi_k^m),\tag{1}$$

where $\Phi_k^m$ is the sampled template for meta-task $m$. Particularly, in tool-augmented scenarios, predicting the output of tools such as QA systems can be impractical. Nonetheless, the retrieval result itself is inherently meaningless (e.g., 'sunny'); however, it gains complete significance when combined with the context of tool calling (e.g., 'the weather in New York is sunny'), as showcased in Figure 3. Thus we modify the context into a more informative state in such scenarios by prompting

LLMs $s_n^* = \text{LLM}(s_n', a_n, t)$, which is trivial for most language models. At last, we arrange multiple metaset pairs of the same toolset into multi-turn QA data as the metaset $\mathcal{M}$, in order to maintain multi-turn dialogue capacity.

### 2.3 TOOL LEARNING AUGMENTED WITH META-TASKS

By incorporating meta-task data, we explore several approaches to augment the tool learning for the purpose of achieving task objectives: a) **In-context learning:** We randomly sample several demonstrations of each meta-task and add them to the system prompt to facilitate tool understanding in a training-free manner. Such task-agnostic knowledge includes the interpretation of rules, supplementing the solution demonstrations. b) **2-stage learning:** Since we aim to build the model's tool understanding as the foundation of tool-oriented learning, an intuitive idea is to train the LLM first on the metasets as the surrogate tasks and then on the solution data $P$. In order to maintain the general ability of the model in the first stage, only the parameters of the query and value projection layers of the Transformer are updated instead of targeting all modules. c) **Data augmentation:** In this approach, we utilize the metaset as the augmented data of conventional instruction tuning methods that the metasets are mixed with solution data and the model is trained uniformly. The model trained on the mixed data is referred to as MetaTool.

## 3 EXPERIMENTS

In this section, we evaluate our approach in both tool-oriented agent and tool-augmented chatbot scenarios. On the one hand, we fine-tune the LLM to master a specific toolset for achieving various complex objectives. On the other hand, we conduct large-scale instruction tuning to enable the model to generalize to new tasks and understand new tools through zero-shot documentation.

### 3.1 TOOL-ORIENTED SCENARIOS

#### 3.1.1 TASK SETUP

We adopt 3 tool-oriented tasks that emphasize complex tool execution and sequential planning. Among them SAW is newly designed while the other two are introduced from PlanBench Valmeekam et al. (2024). The key challenge of these tasks is to understand the rules (preconditions) and the environmental dynamics caused by actions. The task definition and dataset construction are elaborated below.

**SpellAnyWord (SAW).** In this task, the agent needs to sequentially construct a string that contains the target string as a continuous substring. The initial state of the task is a void string. Two non-degradable tools (functions) are avaliable: ***Add***: to add two adjacent letters in the alphabet to the end of the current string. The tool input $\theta$ should be the preceding letter (e.g. passing 'a' to *Add* on current string '' will result in 'ab'). ***Swap***: to swap the position of two adjacent letters in the current string. The input should be the preceding letter (e.g. passing 'a' to *Swap* on 'ab' will result in 'ba'). An example task: The target string is 'any'. A successful action sequence can be [*Add*('a'), *Add*('n'), *Add*('y'), *Swap*('a'), *Add*('o')], which will result in a state sequence ['ab', 'abno', 'abnoyz', 'banoyz', 'banyoz'] and the final string 'b**any**oz' has 'any' as a substring. To eliminate the basis from tokenization, we format each string as a list of letters in prompts throughout the task.

**BlocksWolrd (BW).** In this scenario, the agent needs to stack several blocks on the table into a target state with one hand. Only one block can be moved at a time. Two tools (functions) are avaliable: ***Pick***: to pick a block in the hand. The tool input should be the target block indicated by its color (e.g. *Pick*('yellow')). Blocks cannot be picked if there are blocks on top of them or there's already a block in the hand. ***Stack***: to stack the block in the hand onto the target block or table. The input should be the color of the target block or 'table' (e.g. *Stack*('white'), *Stack*('table')). Blocks cannot be stacked on a block with another block already on top of it or there's no block in the hand.

**Logistics (LOG).** The agent needs to solve a logistics problem by arranging trucks and airplanes to transport the package to the target location. Locations are grouped by cities. Trucks can be used to move packages between locations in the same city and planes can be used to move packages between

| Models | SAW | BW | LOG |
|---|---|---|---|
| ChatGPT | 22.6 | 23.3 | 50.4 |
| ChatGPT-ICL | 20.2 | 20.5 | 43.6 |
| GPT-4 | 28.6 | 43.0 | 46.6 |
| GPT-4-ICL | 27.4 | 40.0 | 37.0 |
| Vicuna-7b | 4.8 | 5.5 | 0.0 |
| LLaMA3-8b-instruct | 6.0 | 6.7 | 6.0 |
| LLaMA3-solution | 9.5 | 19.2 | 8.2 |
| LLaMA3-ICL | 4.8 | 17.8 | 2.0 |
| LLaMA3-2-stage | 9.5 | 21.9 | 12.3 |
| MetaTool (8B) | 32.1 | 37.5 | 30.1 |

Table 1: **Results on tool-oriented tasks.** ICL: in-context learning with meta-task demonstrations. ChatGPT and GPT-4 are provided with tool documentation and few-shot demonstrations.

| E | D | R | I | O | C | S | SAW | BW | LOG |
|---|---|---|---|---|---|---|---|---|---|
| �’ |  |  |  |  |  |  | 9.5 | 27.0 | 11.0 |
|  | ✗ |  |  |  |  |  | 18.5 | 29.0 | 9.0 |
|  |  | ✗ |  |  |  |  | 27.3 | 35.0 | 21.0 |
|  |  |  | ✗ |  |  |  | 17.3 | 32.0 | 18.0 |
|  |  |  |  | ✗ |  |  | 16.1 | 32.0 | 6.0 |
|  |  |  |  |  | ✗ |  | 19.6 | 37.0 | 14.0 |
|  |  |  |  |  |  | ✗ | 15.5 | 31.0 | 10.0 |
| ✓ | ✓ | ✓ | ✓ | ✓ | ✓ | ✓ | 32.1 | 37.5 | 30.1 |

Table 2: **Ablation on tool-oriented tasks.** E: effect meta-set, D: decision-making meta-set, R: Reversion meta-set, I: input boundary meta-set, O: output boundary meta-set, C: counterfact meta-set, S: solution dataset. The crossings denote removing the training data of the corresponding meta-tasks.

cities. Two tools (functions) are available: ***Truck***: to transport the truck and the package (if there is any) from one location to another. ***Plane***: to transport the airplane and the package (if there is any) from one location to another. The tool input should be the starting and ending location indicated by numbers. (e.g. *Truck*(1,2), *Plane*(2,4)). An action is invalid when there is no truck or airplane at the starting location.

**Datasets collection.** For the SAW task, we randomly sample 2k target strings (from 2 letters to 10 letters) as task goals. We modify the BlocksWorld and Logistics tasks from PlanBench into the tool-use version, thus 2k goals for each task are adopted following the original configuration. Optimal action sequences are obtained with heuristic strategy as the solution data.

### 3.1.2 IMPLEMENTATION DETAILS

Our model is fine-tuned based on LLaMA3-8b-instruct AI@Meta (2024) with parameter-efficient fine-tuning method Qlora Dettmers et al. (2024) on 8 A100 GPUs. We utilize the instruction tuning version of LLaMA3 since comprehending tool-oriented tasks with specific objectives is the basis of tool understanding and use. For each task, we train the model on 10k meta-task data and 10k solution data for 3 epochs with AdamW optimizer and the learning rate of 2e-4. The models are tested in a simulated environment that receives the action of using a tool and returns the outcome and current state. We evaluated the model performance on 100 unseen cases of each three tasks.

**Baselines.** We evaluate several baselines to study the effect of different training approaches (illustrated in Figure 1 and described in Section 2.3).(1) LLaMA3-solution: training the base model LLaMA3-8B-instruct merely on the solution data $P$.(2) LLaMA3-ICL: prompting the base model with few-shot meta-task demonstrations (examples shown in Figure 3 and Figure 4).(3) LLaMA3-2-stage: training the base model first on meta-tasks data $M$ then on solution data $P$.

### 3.1.3 RESULTS ANALYSIS

**Overall comparison.** We evaluate the success rate (SR%) of completing each task and show the performances of several models in Table 1. Overall, SOTA closed-source LLMs show impressive zero-shot performance on tool-oriented tasks compared with open-source LLMs including LLaMA3 and Vicuna. By training on both meta-tasks and solution data, our model MetaTool gains significant improvement (+20.9%SR on average) compared with mere training on solution data (LLaMA3-solution). MetaTool also surpasses GPT-4/ChatGPT in the SAW/BW tasks (+3.5%/14.2%SR). ChatGPT represents the model of GPT-3.5-turbo-16k throughout our experiments. Both GPT and LLaMA3 show weaker performances when provided with meta-task demonstrations (ICL) since demonstrating limited cases can be redundant or misleading without proper design. LLaMA3-2-stage that trained on meta-tasks first gains limited improvement compared with the baseline. We conjecture that learning meta-tasks without practicing tool use (training on action sequences) cannot

effectively facilitate tool-use ability with tool understanding. Also fine-tuning with specific QA data may affect the basic linguistic ability of the model. The overall results show that LLMs (including the most powerful ones like GPT-4) still have difficulties conquering complex tool using tasks, especially in planning with tools.

**Ablation study.** We study the ablation of different data components and report the performances in Table 2. It's worth noticing that merely training on meta-tasks can improve the model's zero-shot performance on tool-oriented tasks (line 7), contrary to providing demonstrations of meta-tasks in the system prompt (LLaMA3-ICL in Table 1). When removing QA data from each meta-task, the model performance shows varying degrees of degradation, which verifies the profits of meta-tasks. The meta-tasks of *effect* and *decision-making* have a relatively greater influence on the model's tool understanding capability. Theoretically, these meta-tasks emphasize the causal mechanism of tools that is more fundamental than others.

### 3.2 TOOL-AUGMENTED SCENARIOS

Among the various tool/function calling benchmarks, we study our method on two of the most influential ones: ToolBench Qin et al. (2023c) and Berkeley Function Calling Leaderboard (BFCL) Yan et al. (2024). Training approaches for baselines and implementation details remain the same with tool-oriented scenarios if not specified.

#### 3.2.1 TASK SETUP

**ToolBench** contains diverse user requests with a massive amount (over 16k) of real-world API tools, which are publicly available on the RapidAPI website. The testset is categorized into six distinct groups and contains 1200 instructions (200 each): I1-inst., I1-tool, I1-cat., I2-inst, I2-cat., and I3-inst. Groups labeled with I1, I2, I3 include single-tool tasks, intra-category multi-tool tasks, and extra-category multi-tool tasks, respectively. Groups labeled with "inst.", "tool", "cat." include unseen user instructions, unseen tools, and unseen categories (e.g. sports, entertainment) of tools, respectively. Two evaluation metrics are designed based on ChatGPT: (1) Pass Rate, calculated by the proportion of instructions successfully completed within a limited budget; (2) Win Rate measured by asking a ChatGPT evaluator to select its preference for two solution paths. For each user instruction (e.g. "Can you recommend some popular restaurants within 5km to hold a party?"), the model calls an API and responds to the query based on the tool output.

ToolBench also provides 126k instruction-solution pairs for training, which are generated with GPT4 and depth-first tree search (DFS). GPT-4 gains access to different reasoning paths by choosing either to continue the current node or give up and expand a new node. On top of that, we extract unsupervised tool execution data and generate 650k self-supervised data of meta-tasks following the procedure in Figure 3. We then conduct instruction tuning based on the mixed data and LLaMA3-8B-instruct model, trained for 1 epoch to avoid overfitting. LLaMA3-solution is trained for 2 epochs following the original configuration in ToolBench. Both the solution data and the meta-task data share the same loss setting as we construct the metaset as QA pairs. The context-aware states are generated with the open-sourced LLaMA3-70B-instruct model. All evaluated LLMs are prompted in the ReAct manner to leverage their reasoning ability. Other model implementation details are in line with that described in section 4.1.2.

**BFCL benchmark** is established mainly for the purpose of zero-shot evaluation without holistic training data. The benchmark contains 4251 testing cases in total and is categorized into non-live (self-designed), live (user-contributed), multi-turn, and Hallucination (relevance or irrelevance determination) groups. The model performance is measured by action accuracy with Abstract Syntax Tree (AST) Patil et al. (2023). We test the MetaTool model trained with ToolBench data on BFCL to evaluate the zero-shot ICL ability and generality of our methodology.

#### 3.2.2 RESULTS ANALYSIS

As the ToolBench results shown in Table 3, MetaTool (8B) achieves the second-best performance across all groups merely behind GPT-4, superior to other models including ChatGPT (+8.1% pass rate) and training-based ToolLLaMA-2 (7B) Qin et al. (2023c) (+16.9% pass rate, +8.5% win rate). Especially, our method significantly improves the performance of LLaMA3 (originally incapable)

| Models | I1-Inst. | | I1-Tool | | I1-Cat. | | I2-Inst. | | I2-Cat. | | I3-Inst. | | Averages | |
|---|---|---|---|---|---|---|---|---|---|---|---|---|---|---|
| | Pass↑ | Win↑ | Pass | Win | Pass | Win | Pass | Win | Pass | Win | Pass | Win | Pass | Win |
| ChatGPT | 41.5 | - | 41.0 | - | 41.0 | - | 34.5 | - | 46.5 | - | 22.0 | - | 37.8 | - |
| Claude-2 | 5.5 | 31.0 | 3.5 | 27.8 | 5.5 | 33.8 | 6.0 | 35.0 | 6.0 | 31.5 | 14.0 | 47.5 | 6.8 | 34.4 |
| GPT-4 | **53.5** | **60.0** | **50.0** | **58.8** | **53.5** | **63.5** | **67.0** | **65.8** | **72.0** | **60.3** | **47.0** | **78.0** | **57.2** | **64.4** |
| ToolLLaMA-2 | 25.0 | 45.0 | 29.0 | 42.0 | 33.0 | 47.5 | 30.5 | 50.8 | 31.5 | 41.8 | 25.0 | 55.0 | 29.0 | 47.0 |
| LLaMA3-8B-inst. | 0.0 | 0.0 | 0.0 | 0.0 | 0.1 | 0.0 | 0.0 | 0.0 | 0.1 | 0.1 | 0.0 | 0.0 | 0.0 | 0.0 |
| LLaMA3-2-stage | 31.4 | 43.6 | 35.6 | 44.8 | 40.3 | 44.0 | 40.4 | 48.0 | 36.1 | 46.8 | 28.5 | 58.0 | 34.7 | 47.2 |
| LLaMA3-solution | 32.1 | 45.3 | 39.0 | 43.9 | 36.4 | 43.0 | 40.1 | 52.5 | 40.1 | 43.4 | 35.6 | 61.8 | 37.2 | 48.3 |
| MetaTool (8B) | 42.5 | 52.1 | 41.8 | 51.3 | 43.3 | 46.1 | 52.0 | 54.9 | 50.0 | 54.0 | 45.5 | 74.5 | 45.9 | 55.5 |

Table 3: **ToolBench results.** ChatGPT doesn't have the Win Rate score since all other Win Rates are measured by comparing with its solution paths.

| Models | I1-Inst. | | I1-Tool | | I1-Cat. | | I2-Inst. | | I2-Cat. | | I3-Inst. | | Averages | |
|---|---|---|---|---|---|---|---|---|---|---|---|---|---|---|
| | Pass↑ | Win↑ | Pass | Win | Pass | Win | Pass | Win | Pass | Win | Pass | Win | Pass | Win |
| 2stage-full | 24.8 | 43.0 | 30.0 | 43.9 | 36.0 | 43.0 | 29.2 | 52.1 | 28.9 | 37.7 | 23.7 | 56.8 | 28.5 | 46.1 |
| 2stage-qv | 31.4 | 43.6 | 35.6 | 44.8 | 40.3 | 44.0 | 40.4 | 48.0 | 36.1 | 46.8 | 28.5 | 58.0 | 34.7 | 47.2 |
| solution-1epo | 30.9 | 45.0 | 37.3 | 44.9 | 34.1 | 42.0 | 39.5 | 51.3 | 36.0 | 42.4 | 32.8 | 61.0 | 35.1 | 47.8 |
| solution-2epo | 32.1 | 45.3 | 39.0 | 43.9 | 36.4 | 43.0 | 40.1 | 52.5 | 40.1 | 43.4 | 35.6 | 61.8 | 37.2 | 48.3 |
| MetaTool-1epo | **42.5** | **52.1** | **41.8** | **51.3** | **43.3** | 46.1 | **52.0** | **54.9** | **50.0** | **54.0** | **45.5** | **74.5** | **45.9** | **55.5** |
| MetaTool-2epo | 35.7 | 44.2 | 35.6 | 43.7 | 39.0 | **47.6** | 45.6 | 51.5 | 46.1 | 49.5 | 39.5 | 68.3 | 40.3 | 50.8 |

Table 4: **Ablation results on ToolBench.** '2stage-qv' targets only query and value modules and '2stage-full' targets all parameter modules during the first stage training on metasets. 'solution-*x*epo' denotes baseline LLaMA3-solution trained with *x* epochs.

compared to LLaMA3-solution which is merely trained on solution data (+8.7% pass rate, +7.2% win rate). The comprehensive advantages show the effectiveness of meta-task augmentation. Besides, the notable superiority of GPT-4 can be partly attributed to the fact that all the training and testing instructions are generated with itself. Thus GPT-4 may be more familiar with the distribution and inner motivation of these instructions.

In Table 4, we study the influences of several hyper-settings chosen for our baselines:(1) Comparing with targeting only query and value modules (2stage-qv), The relatively weaker performance of LLaMA3-2stage-full (-2.2%/-1.1% on average) suggests that training on metasets targeting full parameter modules may let the model overfit the QA tasks and hinder the subsequent training on solution data.(2) Results of both LLaMA3-solution and MetaTool trained with 1 epoch or 2 epochs are shown. While early stopping for training merely on solution data harms the performance (-2.1%/-0.5% on average ), early stopping for MetaTool improves the performance (+5.6%/+4.7% on average). The contradiction suggests that training too much on meta-tasks QA data may cause overfitting and weaken the ability of planning actions. Also, learning meta-tasks can bring sufficient knowledge about tools. That helps the LLMs to understand the expert solutions and learn the tool-use tasks faster, thus reduce the need for the second epoch training.

Table 5 shows the zero-shot performance on the BFCL benchmark. It's worth noticing that except for the sets of non-live simple, live simple, and multi-turn base, zero-shot comparison on other test sets is less fair, since MetaTool is merely trained in ToolBench scenarios with a unique task pattern (fixed system prompt) that the model calls a single tool once a time then waits for the tool output. For example, in the "multiple" tasks LLMs are asked to call multiple tools in one response. Nonetheless, we modify the parser of MetaTool to continue generating tokens to fit the requirements of multiple and parallel scenarios. In the "irrelevance" tasks, LLMs have access to tools irrelevant to the instruction and should give up calling any tools, which never occurs in ToolBench scenarios. Therefore in the first place, we count the average accuracy (Simple Ave. in Table 5) of three fair sets and observe that MetaTool surpasses LLaMA3-8B-instruct (+5.3%), LLaMA3-solution (+3.3%) and Hermes-2-theta Teknium (+8.1%) which is also trained based on LLaMA3-8B-instruct, and is close to the latest OpenAI o1-mini (-1.6%). It's also impressive that MetaTool obtains the highest 78.3% accuracy on the non-live simple set, 17.7% higher than the 1st rank model GPT-4-turbo. Despite the

| Models | Non-live | | | | Live | | | | Multi Turn | Hal. | | Simple |
| | simple | multiple | parallel | M&P | simple | multiple | parallel | M&P | base | rel. | irrel. | Ave. |
|---|---|---|---|---|---|---|---|---|---|---|---|---|
| GPT-4-turbo | 60.6 | 91.0 | 90.0 | 89.0 | 67.8 | 74.5 | 75.0 | 62.5 | 33.5 | 70.7 | 79.8 | 54.0 |
| o1-mini | 68.9 | 89.0 | 73.5 | 70.5 | 62.8 | 65.1 | 68.8 | 58.3 | 16.0 | 46.3 | 88.7 | 49.2 |
| Hermes-2 (8B) | 61.3 | 82.5 | 75.5 | 75.0 | 55.8 | 53.1 | 43.8 | 41.7 | 1.5 | 51.2 | 62.7 | 39.5 |
| LLaMA3-8B-inst. | 63.1 | 85.5 | 51.5 | 44 | 60.9 | 60.8 | 37.5 | 20.8 | 3.0 | 75.6 | 27.4 | 42.3 |
| LLaMA3-2-stage | 66.8 | 60.0 | 5.0 | 6.0 | 53.9 | 33.1 | 16.8 | 6.3 | 5.0 | 98.1 | 10.5 | 41.9 |
| LLaMA3-solution | 71.3 | 64.0 | 13.5 | 10.0 | 56.6 | 34.9 | 37.5 | 12.5 | 5.0 | 100.0 | 8.3 | 44.3 |
| MetaTool (8B) | 78.3 | 55.0 | 66.0 | 63.5 | 58.1 | 50.1 | 18.8 | 37.5 | 6.5 | 100.0 | 25.4 | 47.6 |

Table 5: **BFCL results.** M&P denotes the test set of multiple parallel. Hal., rel., and irrel. represent the relevance and irrelevance set of the hallucination group. Simple Ave. denotes the average accuracy of non-live simple, live simple, and multi-turn for a fair comparison. All scores represent the success rates for the test sets.

transferring barriers, MetaTool still achieves moderate performance on the other sets (e.g. multiple, parallel, hallucination) with an average of 52.0% accuracy, significantly higher than the 35.1% of LLaMA3-solution (+16.9%). Overall, the zero-shot results on BFCL clearly demonstrate the exceptional generalizability of MetaTool. With additional data augmentation from diverse scenarios, MetaTool has the potential for significant improvement.

### 3.3 QUALITATIVE CASE OF META-TASKS

As shown in Figure 4, we showcase a qualitative case of meta-tasks data $M$. The tool *search_by_title_for_MDBList* is provided on the real-world API website RapidAPI. The parameters are named casually and we can hardly derive the function of them just by letters (e.g. 's', 'm'). The meta-tasks help the model learn the function and usage of these parameters. For example, from the QA pair of *Effect* meta-task the model observe that feeding 's' as 'friends', 'm' as 'movie', and 'l' as 1 results in a movie titled 'friends'. From the *Input boundary* meta-task, the model learns that 'tv' is not a valid value for parameter 'm'. With multiple QA pairs for each tool, our model is able to learn a more robust tool understanding from actual instances besides descriptions. The tool learning benefits from this paradigm especially in real-world scenarios where the tool descriptions may be diverse and noisy.

## 4 RELATED WORKS

### 4.1 TOOL LEARNING

Recent studies have shed light on the potential of utilizing tools to augment LLMs with external factual knowledge Qin et al. (2023a); Nakano et al. (2021); Song et al. (2023); Hao et al. (2024); Shen et al. (2024); Gao et al. (2023); Wu et al. (2023); Qian et al. (2023); Zhuang et al. (2024); Schick et al. (2024) which is targeted at tool-augmented question-answering scenarios, towards the 'tools for AI' purpose in general. On the other side, With the burgeoning intelligence in reasoning and perception, LLMs' tool-use capability can be widely applied in the automation of various domains including Embodied AI Wang et al. (2024c;b), web manipulation Rawles et al. (2023); Hong et al. (2024); Yang et al. (2023); Deng et al. (2024); He et al. (2024); Zhou et al. (2023), and image/video editing Wang et al. (2024a); Argaw et al. (2022); Hang et al. (2024); Fu et al. (2023).This line of work is intended for tool-oriented planning scenarios for the 'AI for tools' purpose. Effectively mastering complex tools challenges the model to comprehend the precondition and potential outcome of using tools. In this paper, we aim to facilitate LLMs for both tool-oriented and tool-augmented tasks by learning robust tool understanding.

### 4.2 TOOL UNDERSTANDING

As noted by Hernik & Csibra (2009), when learning to utilize a specific tool, children perceive it as an object with particular functions, engaging in a cognitive process to understand its purpose and operation. Analogously, a comprehensive understanding of the tools' functionalities is indispens-

---

**Meta-task demonstrations**

**Tool**: *search_by_title_for_MDBList*. Search for Movies or TV Shows by title, include option year or mediatype filter to narrow your results.
**Parameters**: 1) "s": title to search; 2) "y": year to limit title search in format YYYY; 3) "m": search for "movie" or "tv show". Leave empty for both; 4) "l": limit return items. Use number from 1 to 100.
*Effect*
-Q: When using tool 'search_by_title_for_MDBList' with input {"s": "friends", "m": "movie", "l": 1}, what would be the output/result?
-A: The result would be the search result for the movie titled "friends" limited to 1 item.
*Decision-making*
-Q: What function should be called and what parameters should be fed into it to get "the top-rated movies from around the year 1990, limited to three results."? Examples: "The Unforgettable Director of Love Movies", "Killed the Family and Went to the Movies", ...
-A: Tool: search_by_title_for_MDBList. Inputs: {"s": "the top-rated movies", "y": 1990, "m": "movie", "l": 3}
*Input boundary*
-Q: Would {"s": "family", "m": "tv"} be considered a suitable input for tool search_by_title_for_MDBList?
-A: No, since "tv" is invalid for parameter "m".
*Output boundary*
-Q: Is it feasible to get the movies titled by "friends" in Singapore as the result with tool search_by_title_for_MDBList?
-A: No, it's infeasible since it's beyond the capability of the tool.
*Counterfact*
-Q: When using tool 'search_by_title_for_MDBList' with input {"s": "friends", "m": "movie", "l": 1}, I get the movie titled "friends" limited to 1 item as the output. What would be the new output if I change the input "l" as 5?
-A: The new tool output is:the movie titled "friends" limited to 5 items.

Figure 4: Qualitative case for meta-task generation. Given a real-world tool (API), we obtain a set of QA pairs for 6 meta-tasks following the 2-step procedure illustrated in Figure 3.

able for enabling the controller to use tools proficiently. In real-world scenarios, tools are typically accompanied by a manual (or tutorial), which provides sufficient relevant details about their functionalities and usage. Endowed with strong few-shot learning Brown et al. (2020) and zero-shot learning Wei et al. (2021) capabilities, foundation models can be prompted to unravel tools' functionalities and comprehend how to use them. To this end, we can construct suitable task-specific prompts either through manual design Vemprala et al. (2024) or retrieval Zhou et al. (2022). However, prompting is restricted by input context length, thus the situation may be more challenging with multiple complex tools with long descriptions. While most training-based tool learning methods rely on extensive expert-annotated solution data for goal-oriented tasks, the knowledge contained in the tool execution process itself remains unutilized. We propose a self-supervised data augmentation method to efficiently endow LLMs the comprehension of a set of tools.

## 5 CONCLUSION

In this work, we introduced MetaTool, an efficient and generalizable method that facilitates tool learning with task-agnostic comprehension. This is achieved by deriving self-supervised meta-task data from tool execution actions. Augmented the meta-tasks into either complex toolset fine-tuning or large-scale instruction tuning, our model exhibits sophisticated tool mastery as well as generality in in-context learning. Evaluated on multiple tool use benchmarks, MetaTool outperforms models trained on expert solutions and showcases comparable performance with ChatGPT in a size of 8B.

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

# A    APPENDIX

## A.1    PROMPTS

---

**Context Generation Prompt**

**System**
You are a helpful assistant. You will first be provided with the documentation of several tools and their functions/APIs. Then, given the called function, the function input, and the tool output/result from the user, your task is to provide the context that represents the output.
**Instructions**
1. Review the provided tool documentation to understand the available functions and their purposes.
2. Identify the called function and its input parameters.
3. Describe what context the result represents based on the function and input.
**Examples**
**Tool documentation.** Tool: Weather; Function/API: *get_weather*; Parameters: *city* (string), *data* (string)
**Input.** Called function: *get_weather*; Function input: {'city': 'New York', 'data': 'July 10th'}; Result: 'sunny'
**Output.** The weather in New York on July 10th.
**Notice**
-Be precise and DO NOT include the details in the result.

---

Figure 5: Prompt for generating contextual result given the tool description, action, and the original action result.

---

**Self-play Trial Prompt**

You are AutoGPT, you can use many tools(functions) to do the following task.
First, I will give you the task description, and your task start.
At each step, you need to give your thought to analyze the status now and what to do next, with a function call to actually execute your step. Your output should follow this format:
Thought:Action:Action Input:
After the call, you will get the call result, and you are now in a new state.
Then you will analyze your status now, then decide what to do next...
After many (Thought-call) pairs, you finally perform the task, and then you can give your final answer.
**Remember:**
1. the state change is irreversible, you can't go back to one of the former states. If you want to restart the task, say "I give up and restart".
2. All the thought is short, at most in 5 sentence.
3.You can do more than one try, so if your plan is to continuously try some conditions, you can do one of the conditions per try.
Let's Begin!
**Task description:** You should use functions to help handle real-time user queries. Remember:
1.ALWAYS call the "Finish" function at the end of the task. The final answer should contain enough information to show to the user. If you can't handle the task, or you find that function calls always fail(the function is not valid now), use function Finish("give up and restart").
2.Do not use origin tool names, use only subfunctions' names.
You have access to the following tools: (tool descriptions)
{'name': 'Finish', 'description': 'If you believe that you have obtained a result that can answer the task, please call this function to provide the final answer.'}

---

Figure 6: Prompt for search for solution paths through self-play trials. The successful trials are picked according to the final answer. Unsupervised tool actions can be extracted from all trials.

## A.2 QUALITATIVE CASE STUDY

---

**Context Generation Examples**

**Example 1**
**Input for LLMs**: Tool: *fixtures_for_golf_leaderboard* (Lists tournament fixtures for a given tour_id and season_id).
Input parameters: {"tour_id": 1, "season_id": 2023}.
Result: "2023 European Tour"
**Output (contextual result)**: Golf fixture held in 2023 season with tour_id 1 is 2023 European Tour.

**Example 2**
**Input for LLMs**:
Tool: *find_place_by_text* (Search for places by text string).
Input parameters: {"text": "national parks in United States"}.
Result: {'name': 'Yellowstone National Park', 'address': 'United States', 'phone_number': '+13073447381', 'website': 'https://www.nps.gov/yell/index.htm'}
**Output (contextual result)**: Details of national parks in the United States such as Yellowstone National Park.

**Example 3**
**Input for LLMs**:
Tool: *get_10000_anime_quotes* (Access best quality anime quotes from over 10000+ animes. Get anime quotes based on anime search.)
Input parameters: {"search": "Naruto", "page": 1}
Result: ['Those who do not understand true pain can never understand true peace.', 'Things never go as planned in this curse world.', 'Becoming the Hokage doesn't mean people will acknowledge you.']
**Output (contextual result)**: A list of quotes from the anime "Naruto" retrieved from page 1 including 'Things never go as planned in this curse world'.

---

Figure 7: Examples of generating contextual results with LLMs given the tool descriptions, actions and action results.

We present several examples of context generation (mentioned in Section 2.2) and the prompt in Figure 7 and Figure 5. The LLM worker is only asked to complete the contextual information for the results returned by the tool. In the Effect meta-task, the model learns to predict the contextual results given the input parameters, which helps it better understand the tool mechanism. Otherwise asking the model to predict merely the retrieval results (e.g. 2023 European Tour, Yellowstone National Park) is impractical and not beneficial. No other information or prior knowledge from the LLM worker are needed and provided to MetaTool.

As showcased in Figure 8, the agent is required to construct stacks containing a green block on top of a yellow block from a pile of 4 blocks. With mere descriptions of tools in the prompts, LLaMA3 fails to understand the precondition of using tools resulting in invalid actions. Training on tool-oriented solution data, LLaMA3-solution attempts to lift the yellow block successfully but fails to sequentially achieve the task goal and falls into repetitive loops. The proposed MetaTool model achieves the target state with an effective action sequence (although still not the optimal efficiency) and corresponding reasoning. These 3 models correspond to the 3 paradigms illustrated in Figure 1. The results show that LLMs can learn tool use better on the basis of robust tool understanding.

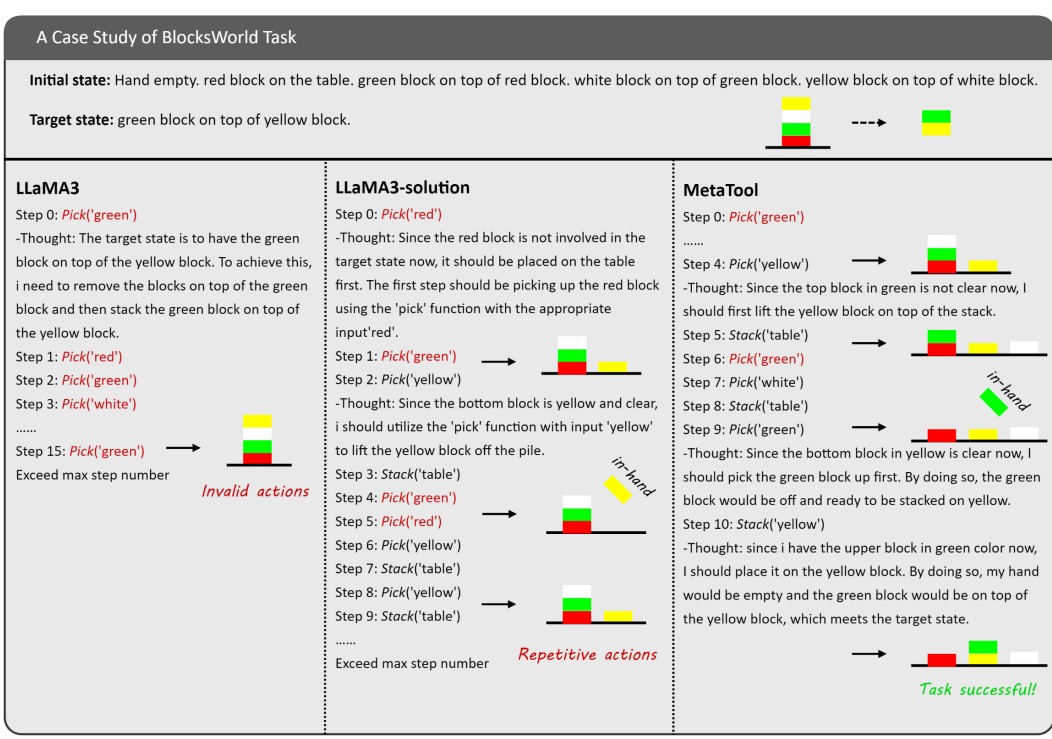

Figure 8: Case study of MetaTool compared with 2 baselines on BlocksWorld task. Actions in red denote invalid ones (e.g. pick up a block at the bottom). LLaMA3-solution is the LLaMA model trained on task solution data.

Meta-task demonstrations:

- *Effect*

-**Q**: The current text is 'uv'. When using tool 'Add' with input 'd', what would be the output/result?

-**A**: The result would be 'uvde'.

......

- *Decision-making*

-**Q**: If the original text is 'uvde' and the desired outcome using the 'Swap' is 'vude', what is the necessary input for this function?

-**A**: The tool input is 'u'.

......

- *Input*

-**Q**: Given the existing text 'uv', would 'hi' be considered a suitable input for the function 'Add'?

-**A**: No, it's invalid since 'Add' only takes one letter as the input.

......

- *Output*

-**Q**: The current text is 'uvde'. Is it feasible to get 'dvue' as the result with the tool 'Swap'?

-**A**: No, it's infeasible since 'Swap' can only change the positions of two letters once.

......

- *Counterfact*

-**Q**: When using tool 'Add' with input 'd', I get 'uvde' as the output. What would be the new output if I change the input as 'e'?

-**A**: The new tool output is: 'uvef'.

......

Task-oriented solutions:

-User: target text: 'word'

-Assistant: Thought: The current text '' does not contain all the letters in the target text. So I should...

Action: Add Action input: 'v'.

-Function: 'vw'    *step 0*

......

-Assistant:Thought:...Action: Add Action input: 'd'

-Function: 'vwoprsde'    *step 3*

-Assistant: Thought: Although the current text contain all the letters in the target text, the order of those letters does not match. So I should... Action: Swap Action input: 'p'

-Function: 'vworpsde'    *step 4*

......

-Function: 'vwordpse'    *step 6*

*task successful !*

Figure 9: Qualitative cases for meta-tasks in tool-oriented scenario.

