# OpenReview forum: "MetaTool: Facilitating Large Language Models to Master Tools with Meta-task Augmentation"
_ICLR.cc/2025/Conference — Submitted to ICLR 2025_

### Official Review · Reviewer_vUAy · 2024-11-02

**Soundness:** 3
**Presentation:** 2
**Contribution:** 2
**Rating:** 6
**Confidence:** 4

**Summary:**

This paper introduces MetaTool, a methodology for improving how large language models (LLMs) learn to use tools. Instead of relying solely on few-shot prompting or supervised fine-tuning with expert annotations, the authors propose a self-supervised approach based on meta-tasks that capture fundamental aspects of tool usage. The method generates training data by predicting masked elements in tool execution processes, enabling LLMs to develop a deeper understanding of tool functionality, causality, and constraints. The approach demonstrates improvements in both tool-based planning and chatting scenarios, achieving performance comparable to ChatGPT while using much smaller models.

**Strengths:**

- The data generation process is scalable: The paper uses self-supervised techniques to generate training data for tool use via a list of meta-tasks without requiring expert annotations. The meta-task framework allows for the automatic generation of diverse training examples that cover various aspects of tool understanding.

- Reasonable Meta-Task Design: The six meta-tasks (Effect, Decision-making, Reversion, Input Boundary, Output Boundary, and Counterfact) are well-designed to capture different aspects of tool understanding and/or action execution, which seems generalizable.

- Good empirical results: The experimental results show decent performance gains, with MetaTool achieving comparable results to ChatGPT while using much smaller models (8B parameters). Also ablation study on tool-oriented tasks showcase the effectiveness of meta-tasks generation even in the absence of “solution” data. Ablation study in ToolBench and BFCL validate the effectiveness of this data generation approach.

**Weaknesses:**

- Lack of clarity regarding “solution data”: It feels like the author didn’t fully introduce and define these -concepts and brought them up abruptly in line 227. And in the evaluation section, it is a little unclear what data each baselines were trained on. It would be helpful if the authors could define different types of data in the experiment setup and use a consistent notation across the entire paper.

- Lake of details about the “meta-task generation”: it would be good to replace figure 4 with a qualitative example of metatasks generated for real-world benchmarks (e.g., ToolBench, BFCL). Also I’d appreciate the authors providing the actual prompt used for metaset construction, specifically about L199-201 “For large toolsets and diverse task scenarios that are hard to enumerate, we incorporate LLMs with self-play or tree search techniques to reduce redundant trials...”

**Questions:**

- The authors mentioned, “we modify the context into a more informative state in such scenarios by prompting LLMs s ∗ n = LLM(s ′ n , an, t), which is trivial for most language models.” Will the complexity of this generated instruction have an impact on the model performance? It would be helpful to see a few qualitative examples of this "context generation" (e.g., prompt input, output).

---

> ### Author Response · Authors · 2024-11-21
> **Rebuttal with clarification and revised paper**
>
> We thank the reviewer for the affirmative evaluation and the constructive suggestions on improving our work. We will reply to the concerns in detail in the following lines.
>
> Q1. **Lack of clarity regarding “solution data”.** The reviewer points out that the concept of solution data is not fully introduced and suggests "defining different types of data in the experiment setup and using a consistent notation across the entire paper."
>
> A1. **A sample of solution data (or a solution path) mentioned in our paper can be defined as a sequence of actions and states** $p=${$s_1, a_1, ..., s_T, a_T$}$\in P$, where $T$ is the number of steps to reach the terminal state. The solution path should lead to an objective state or satisfy the user's instruction. Note that in practice each action should include a "thought" before calling the tool with inputs, in order to elicit the reasoning ability of LLMs. For clarity, we now describe the data each baseline is trained on: LLaMA3-solution and ToolLLaMA are trained merely on the solution data $P$. MetaTool and LLaMA3-2-stage are trained with both the solution data $P$ and the meta-tasks data $M$. We have revised our paper to include the formalization (line 134) and clarification (Section 3.1.2 lines 307-311) above and kept consistent notation throughout the paper.
>
> Q2. **Lack of details about the “meta-task generation”.** The reviewer kindly offers suggestions for a better demonstration of our work including "replace figure 4 with a qualitative example of meta tasks generated for real-world benchmarks" and "providing the actual prompt used for metaset construction".
>
> A2. Thanks for the suggestion to improve our demonstration. We have added a new figure in our revised paper (Figure 4 in lines 486-514) showcasing the meta-tasks generated for ToolBench. The tool *search_by_title_for_MDBList* is provided on the real-world API website RapidAPI (https://rapidapi.com/hub). The parameters are named casually and we can hardly derive their function just by letters (e.g. ’s’, ’m’). The meta-tasks help the model learn the function and usage of these parameters. For example, from the QA pair of Effect meta-task the model observe that feeding ’s’ as ’friends’, ’m’ as ’movie’, and ’l’ as 1 results in a movie titled ’friends’. From the Input boundary meta-task, the model learns that ’tv’ is not a valid value for parameter ’m’. With multiple QA pairs for each tool, our model is able to learn a more robust tool understanding from actual instances besides descriptions. The tool learning benefits from this paradigm especially in real-world scenarios where the tool descriptions may be diverse and noisy.
>
> We also provide the actual prompt for both context generation and solution path searching in Appendix A.1 (lines 706-755). Please feel free to check them out and let us know if there is any issue.
>
> Q3. **Qualitative examples of context generation.** The reviewer is concerned about the impact of the complexity of the generated contextual instruction and calls for "a few qualitative examples of this context generation".
>
> A3. We understand that the lack of examples for generating contextual instructions has led to ambiguity and concern. Here is a qualitative example of this process and we have also included more in Appendix A.2 (lines 758-788):
>
> **Input for LLMs:** Tool: fixtures_for_golf_leaderboard (Lists tournament fixtures for a given tour_id and season_id). Input parameters: {"tour_id": 1, "season_id": 2023}. Result: "2023 European Tour"
>
> **Output (contextual result):** Golf fixture held in 2023 season with tour_id 1 is 2023 European Tour.
>
> As shown above (prompt provided in Appendix A.1), the LLM worker is only asked to complete the contextual information for the results returned by the tool. In the Effect meta-task, the model learns to predict the contextual results given the input parameters, which helps it better understand the tool mechanism. Otherwise asking the model to predict merely the retrieval results (e.g. 2023 European Tour) is impractical and not beneficial. No other information or prior knowledge from the LLM work is provided.

---

> ### Author Response · Authors · 2024-11-28
>
> Dear reviewer vUAy:
>
> I hope this message find you well. We have carefully considered your feedback and have made corresponding improvements to the manuscript. We truly value your insights, and your expertise has greatly contributed to enhancing the quality of our work. Could you please let us know if the revisions meet your expectations? As the deadline for discussion nears, we kindly ask if you could review our rebuttal and updated paper. We are eager to address any additional questions that may arise. Thank you for your valuable support and consideration.
>
> Sincerely,
>
> Authors

---

> > ### Comment · Reviewer_vUAy · 2024-12-01
> >
> > I appreciate the authors' response. My concerns are addressed and I will keep my score.

---

### Official Review · Reviewer_BBkH · 2024-11-02

**Soundness:** 3
**Presentation:** 3
**Contribution:** 3
**Rating:** 6
**Confidence:** 3

**Summary:**

This paper introduces MetaTool, a new way to help large language models (LLMs) better understand and use tools. Instead of relying on traditional methods, like giving examples in prompts or using labeled training data, MetaTool focuses on self-supervised learning through meta-tasks. The idea is to teach models the basics of how tools work, like understanding cause and effect, what actions are allowed, and what outcomes to expect. They introduce six meta-tasks—Effect, Decision-making, Reversion, Input Boundary, Output Boundary, and Counterfact—that cover these foundational ideas. MetaTool shows impressive results across different tool-based tasks and even competes well against models like ChatGPT in both planning and chat scenarios.

**Strengths:**

+ The approach is unique because it emphasizes a general, task-independent understanding of tools. MetaTool's focus on foundational tool knowledge is different from the usual heavy reliance on labeled data, allowing the model to handle more situations with less specific training.

+ The experiments are thorough, covering both scenarios where the model has to use a tool in sequence (like planning) and where it’s just one part of a conversation or task. MetaTool consistently performs well across the board, and the ablation study is detailed, showing which parts of the model contribute the most to performance.

+ The explanation of the meta-tasks is clear and easy to follow. Each task seems thoughtfully designed to address different aspects of tool use, and the visuals in the paper, like the figures comparing MetaTool with other methods, really help make the results easy to understand.

**Weaknesses:**

- While the benchmarks are solid, some of them are in simulated environments. This setup might not fully reflect the unpredictability of real-world situations, so testing MetaTool in live, dynamic environments could strengthen the case for its broader applicability.

- The self-supervised meta-task setup is definitely innovative, but it might be tricky for people who want to apply it to new tools or unique domains without more support. Offering some guidance or a framework for adding new tools could make MetaTool easier to use widely.

- Although MetaTool performs well against strong baselines, it would be interesting to see it compared to other recent multi-task or hierarchical learning approaches, which also aim to improve model generalization. This would give a better sense of where MetaTool stands in terms of scalability and flexibility.

**Questions:**

+ Could MetaTool work with tools in more dynamic, unpredictable domains, like finance or autonomous driving, where tool results might vary or need to adapt in real-time?

+ The paper mentions that MetaTool’s data generation is efficient, but how does this scale up if we’re working with very large toolsets? What’s the computational cost?

+ Do the authors plan to add more meta-tasks or tweak existing ones to improve model understanding? Would expanding to include tasks around probabilistic reasoning or continuous learning help MetaTool become even more generalizable?

---

> ### Author Response · Authors · 2024-11-21
> **Rebuttal with clarification and paper revision**
>
> Thanks for the detailed questions and constructive suggestions for our work. We will reply to them in detail as follows.
>
> Q1. **Testing in live and dynamic domains.** The reviewer is concerned that the experimental setup "might not fully reflect the unpredictability of real-world situations" and suggests that we should "test MetaTool in live and dynamic environments like finance or autonomous driving".
>
> A1. **While we apply simulated environments for the tool-oriented scenario, real-world APIs and live user queries are used to evaluate our method for the tool-augmented scenario.** On one side, given the nature of the tool-oriented scenario that the model needs to observe the new environmental state after each action, it's more feasible and reproducible to develop a self-host environment (e.g. BW and LOG in PlanBench and some tasks in BFCL). Still, it's indeed important to expand the study from simulated environments to real-world environments. And we are looking forward to advancing this process forward. On the other side, it's worth noticing that in the tool-augmented chatbot scenario, over 16k real-world APIs are included in ToolBench and they are connected to the live and dynamic internet such as Instagram and YouTube. BFCL also considers live tasks that the instructions are contributed by real-world users. Thus, the results and evaluations on these benchmarks substantiated the applicability of MetaTool in similar scenarios.
>
> Q2. **Guidance for adding new tools and scaling up.** The reviewer is concerned that "it might be tricky for people who want to apply it to new tools or unique domains without more support" and suggests "Offering some guidance or a framework for adding new tools". The reviewer also asks "How does this scale up if we’re working with very large toolsets? What’s the computational cost?"
>
> A2. **Our approach can be easily transferred to new toolsets in two optional manners:** (1) Through instruction tuning on large-scale datasets with a large number of tools (similar to us training MetaTool on ToolBench), the model gains zero-shot generalizability to understand and use new tools according to their documentations. Since our method does not require human annotation, the data synthesis process is easy to scale up by filling actions or states into QA templates. (2) Generate data for new tools or new domains and train a model to master the tools. Here is a specific step-wise guide for adapting new tools in new domains: First, determine if there's solution data that includes successful paths of actions and results. If not, an easy way is to prompt advanced LLMs to trial multiple times (example prompts are showcased in Appendix A.1 in lines 728-755) and pick the paths with successful final results. Second, extract unsupervised tool-use data samples, each of which contains the tuple of action $a$, initial state $s$, and new state $s'$. Third, synthesize the self-supervised meta-task data for each sample by filling the variables into QA templates (showcased in Figure 4). Fourth, augment the solution data with the meta-task data and train the base model through supervised fine-tuning. We will also release an operable codebase to guide this practice.
>
> Q3. **Comparison to multi-task or hierarchical learning approaches.** The reviewer is curious about the scalability and flexibility of our method "compared to other recent multi-task or hierarchical learning approaches, which also aim to improve model generalization"
>
> A3. One of the most crucial challenges of tool learning is tool/task generalization. We believe multi-task learning or learning with a hierarchical framework has great potential for application in tool learning and are glad to discuss the comparison between them and our method. To address your concern more effectively, could you please specify the particular methods or papers you are referring to? We are looking forward to further discussion.
>
> Q4. **Future explorations for MetaTool.** The reviewer asks about improving model understanding with more or modified meta-tasks and discusses the potential of "including tasks around probabilistic reasoning or continuous learning to help MetaTool become even more generalizable".
>
> A4. Thanks for the inspirational idea. **Developing more task-agnostic meta-tasks can be a promising exploration direction to improve tool understanding and generalization.** The motivation behind our meta-tasks is to provide tool knowledge that is transferable across various tasks. A comprehensive set of meta-tasks is defined asking the model to predict each key element (e.g. actions, states, boundaries) of the tool-use process. Besides those fundamental elements, other tool knowledge can also be useful regarding different tools and scenarios. For example, we can ask the model to predict the probability of successfully changing the environmental state facing unpredictable domains and annotate the answer through repeated sampling and analyzing the results.

---

> > ### Comment · Reviewer_BBkH · 2024-11-25
> > **Response**
> >
> > Thank you for your response

---

> > > ### Comment · Reviewer_BBkH · 2024-11-28
> > > **Update of score**
> > >
> > > The authors addressed some of my concerns, hence I've raised my score from 5 to 6.

---

> > > > ### Author Response · Authors · 2024-11-29
> > > > **Additional results and examples**
> > > >
> > > > Thank you very much for reading our rebuttal and raising the score.
> > > >
> > > > We understand that our previous explanation of the experiment in live and dynamic domains may not have been specific enough. Therefore, we are providing additional results and examples specifically focused on live and unpredictable domains below. We selected two tool categories (`Finance` and `Media`) from ToolBench and tested the models using subsets of ToolBench that involve these tools and their corresponding instructions. These tools (APIs) are connected to the real-world internet and return dynamic and unpredictable results over different periods. Below, we showcase some of the tools and queries our model was tested with:
> > > >
> > > > **1. Financial Tools**:
> > > > * *Commodity Groups* (Retrieve data for commodity groups. Source page: https://www.investing.com/commodities)
> > > > * *Metals Futures Prices* (Retrieve data for metals prices by date)
> > > >
> > > > **Instruction**:
> > > > I am a financial consultant and I need real-time data on commodities futures prices. Can you provide me with the latest quotes for metals? Additionally, I would like to know the commodity groups these futures belong to.
> > > >
> > > > **2. Media Tools**:
> > > > * *GetVideosByTag for Vimeo* (Retrieve a list of videos that have the specified tag. Source page: https://vimeo.com/)
> > > > * *SearchVideos for Vimeo* (Search for videos according to the format and query.)
> > > >
> > > > **Instruction**:
> > > > I'm a film student conducting research on videos with the tag 'animation'. Can you provide me with videos that have this tag? I would like to see the most commented videos first.
> > > >
> > > > Below, we present the quantitative results in the table. We tested the models on 48 tasks in the `Finance` category and 47 tasks in the `Media` category, involving a total of 259 different tools. As shown in the table, our model MetaTool achieves the best performance in the `Finance` domain and is closely behind GPT-4 in the `Media` domain. MetaTool also shows significant improvement compared to the LLaMA3-solution baseline (+32.4/+18.3 points on average), which was trained solely on solution data. This domain-specific study further verifies the generalizability of our method when facing dynamic and unpredictable environmental feedback.
> > > >
> > > > | Models          | Finance   |          | Media     |           |
> > > > |-----------------|-----------|----------|-----------|-----------|
> > > > |                 | Pass Rate | Win Rate | Pass Rate |  Win Rate |
> > > > | ChatGPT         | 68.8      | -        | 23.4      | -         |
> > > > | GPT-4           | 66.7      | 52.1     | **48.9**  | **62.8**  |
> > > > | ToolLLaMA-2     | 25.0      | 29.2     | 4.3       | 21.4      |
> > > > | LLaMA3-solution | 14.6      | 22.9     | 40.4      | 58.5      |
> > > > | MetaTool        | **75.0**  | **56.3** | 44.7      | 61.7      |
> > > >
> > > > Once again, thank you for taking the time and effort to review our manuscript. If there are any remaining concerns about our work, please let us know, and we will be happy to address them and improve our paper.

---

### Official Review · Reviewer_oF9y · 2024-11-04

**Soundness:** 2
**Presentation:** 1
**Contribution:** 3
**Rating:** 5
**Confidence:** 4

**Summary:**

This paper proposes MetaTool, a novel approach for training Large Language Models (LLMs) to use tools effectively. The authors argue that current methods, which rely on demonstrations or expert annotations, need to be revised in generalizing to complex tools and tasks. MetaTool addresses this by introducing a set of self-supervised meta-tasks that focus on the fundamental nature of tools, such as causality and constraints. These meta-tasks are used to generate high-quality training data without human annotation. Through extensive experiments, MetaTool demonstrates superior performance to other LLMs on tool-oriented tasks, achieving results comparable to ChatGPT, and exhibits impressive generalization capabilities on new tasks.

**Strengths:**

This is a paper with good methodology and solid experiments.

The paper tackles the crucial challenge of enabling LLMs to effectively use tools, a capability essential for expanding their real-world applicability.

The paper demonstrates a good understanding of related LLMs and tool learning work, with a comprehensive list of references.

 The experimental setup and evaluation metrics are well-defined, the ablation studies are comprehensive, and the results demonstrate a clear improvement over baselines trained solely on solution data.

The tool-oriented scenario is an interesting setup. This is a new environment to evaluate LLMs' tool-using capability.

**Weaknesses:**

This paper has several weaknesses regarding the writing and the experiment results.

The idea of using self-supervised tasks to improve tool understanding is not entirely new. The authors themselves acknowledge that previous works like Toolformer and TALM have explored similar approaches. The main difference in MetaTool seems to be the specific set of meta-tasks proposed, but their novelty and contribution need further justification.

 The problem formulation and definition are clear. However, I am also curious about the real-world scenarios of these six settings. Are there any specific rationales behind why we should apply this causal inference mechanism to tool use?

 The paper lacks a comprehensive comparison with a wider range of state-of-the-art tool-learning models, including those employing self-supervised learning. This limitation makes it difficult to claim that MetaTool significantly improves over existing approaches [1]. There are more baselines on the Berkeley Function-Calling Leaderboard to compare with. Some of them are trained from LLaMA-3.1-8B as well.

 From what I can read through this paper, the self-play and tree search lack implementation details. This is an interesting way of generating synthetic data. However, the authors did not explain that clearly. For example, the method section should describe using the ReAct to generate thought-tool-input tuples instead of the experiment section.

Some claims or writings in this paper are contradictory or confusing.

 The analogy to BERT isn't that pertinent to me. We are still training the LLMs under an autoregressive objective with LLaMA-3-8B. The model is trained on an augmented/generated dataset using BERT's 'masking' process but is not trained to predict what is missing in the context.

What is the model version of ChatGPT? What about GPT-4 and Claude-2? The authors should specify the model versions more clearly.

 It will be helpful if the authors add $\uparrow$ and $\downarrow$ to show which directions of the metrics are better.

 Line 225 ... are updated instead of full-parameter training. Which part of the training requires full-parameter training? Isn't this model trained using QLoRA?

 The tables and figures are organized confusingly. Table 3 is for tool-augmented scenarios but is placed in Section 3.1. Figure 4 is for tool-oriented scenarios, but the authors separate Section 3.3 for result analysis (only for the results in Section 3.1).

 All the tables and figures should have hyperlinks.

**Questions:**

Typo

Line 146: Bert $\rightarrow$ BERT

Line 208: ',' should follow the math formula

Line 256: BlocksWolrd $\rightarrow$ BlocksWorld

Line 334: ChaGPT $\rightarrow$ ChatGPT

Line 363: ReACT $\rightarrow$ ReAct

[1] Devlin, J. (2018). BERT: Pre-training of deep bidirectional transformers for language understanding. arXiv preprint arXiv:1810.04805.

[2] OpenAI (2022). Introducing ChatGPT. https://openai.com/index/chatgpt.

[3] Yao, S., Zhao, J., Yu, D., Du, N., Shafran, I., Narasimhan, K. R., \& Cao, Y. ReAct: Synergizing Reasoning and Acting in Language Models. In The Eleventh International Conference on Learning Representations.

Clarification

Line 352: Two evaluation metrics are designed based on ChatGPT: (1) Pass Rate, calculated
by the proportion of instructions successfully completed within a limited budget; (2)Win Rate,
measured by asking a ChatGPT evaluator to select its preference for two solution paths.

Question: What does 'based on ChatGPT' mean? If you use the Pass Rate, it is an automatic evaluation that does not require ChatGPT.

---

> ### Author Response · Authors · 2024-11-21
> **Rebuttal with clarification and paper revision**
>
> We thank the reviewer for the constructive concerns. We address them in detail in the following lines.
>
> Q1. **Real-world rationales behind 6 meta-tasks.** The reviewer asks about the "real-world scenarios of the six settings of meta-tasks" and "specific rationales behind why we should apply this causal inference mechanism to tool use."
>
> A1. **The rationale for designing meta-tasks based on the causal theory is that there naturally exists a cause-effect relation between the tool use and its outcome.**  Regarding the tool-use process as a state transition (as formally defined in section 2.1), the causality can be denoted as $A \rightarrow S' \leftarrow S$, where the arrows represent the causal influences. In particular, action $A$ is an intervention in which we actively affect the state by using a tool, rather than observing the correlation between actions and outcomes, which is theoretically introduced in [5]. Understanding those causalities helps the model understand the tool mechanism better. For example in a real-world scenario (from the BFCL benchmark), taking the `mv` command in Linux (mv(<original file>, <target directory>)) as a tool, the action of using it actively changes the state of the file system. Learning meta-tasks such as "What's the outcome of calling tool `mv` with parameters of 'test.py' and '/home/codes/'?" and "What parameters should you pass to tool `mv` to move the 'test.py' file to directory '/home/codes'?" let the LLM understand the tool mechanism and how to actively achieve a desired outcome. We update more examples of meta-tasks in real-world scenarios in Figure 4 of our revised paper. Please feel free to browse them.
>
> Q2. **Comparison with other tool-learning methods.** The reviewer suggests that "the novelty and contribution need further justification" and calls for "a comprehensive comparison with a wider range of state-of-the-art tool-learning models including those employing self-supervised learning" and "more baselines on the Berkeley Function-Calling Leaderboard to compare with. Some of them are trained from LLaMA-3.1-8B as well."
>
> A2. We appreciate the concern for a more comprehensive comparison and now further elaborate on our contribution. **Firstly, although methods like Toolformer[1] and TALM[2] also adopt the idea of self-supervised learning and do not require extra human annotation, the method we propose is essentially distinct from them.** Toolformer embeds the successful tool actions and their results in the model's answer to emphasize 'when' to use tools during question answering. TALM employs an iterative self-play technique to collect successful solutions by exploring with LLMs themselves. Both methods are task-related in that they depend on a stable judgment of whether the task is successful and merely utilize valid actions. On the contrary, MetaTool aims to excavate casual knowledge of tools that exists and is transferable in various tasks, which is theoretically supported by [6]. We also make use of failed tool-use experiences such as the invalid input cases in the input boundary meta-task. This distinction enables us to apply and validate our method in various scenarios.
>
> **Secondly, the tasks and tools those methods are tested on are dramatically different from ours.** While Toolformer is designed and evaluated considering several typical tools (e.g. QA language model, Wikipedia search, canlendar) and TALM considers only a text-to-text API for QA tasks in two domains, we evaluated MetaTool with over 16k tools for both multi-step planning and multi-round chatbot scenarios. Given that both of them are not officially open-sourced, it's impractical to reproduce them and transfer them to our tasks while ensuring a fair comparison with their original implementations.
>
> **Thirdly, the experiment on BFCL undoubtedly demonstrates the zero-shot generalizability of MetaTool.** Essentially, we evaluate our methods on BFCL to answer the question: "Can our self-supervised method enable the zero-shot generalization to new tasks and environments?" Comparison with the LLaMA3-solution and LLaMA3-8B-instruct (shown in the table below) verifies the effectiveness of MetaTool in enhancing generalizability. Although there are other models on the BFCL, they don't share the same base model with MetaTool (i.e. LLaMA3-8B-instruct) and most haven't released their methods and datasets for tool learning, which makes the comparison less meaningful and less fair. Also, comparing with models trained from LLaMA-3.1-8B would be unfair since LLaMA3.1 has been trained for tool use based on LLaMA3. Thus we showed the performance of GPT-4-turbo (top-1), o1-mini, and Hermes-2 (the one trained based on LLaMA-3-8B) to show the relative capability of our model as reference.

---

> ### Author Response · Authors · 2024-11-21
>
> |                    | nonlive-AST |          |          |      | live-AST |          |          |      | Multi-turn | Hal.  |        | Ave. |
> |:------------------:|:-----------:|:--------:|:--------:|:----:|:--------:|:--------:|:--------:|:----:|:----------:|:-----:|:------:|:----:|
> |                    | simple      | multiple | parallel | M&P  | simple   | multiple | parallel | M&P  | base       | rel.  | irrel. |      |
> | LLaMA3-8B-instruct | 63.1        | 85.5     | 51.5     | 44   | 60.9     | 60.8     | 37.5     | 20.8 | 3          | 75.6  | 27.4   | 42.3 |
> | MetaTool           | 78.3        | 55.0     | 66.0     | 63.5 | 58.1     | 50.1     | 18.8     | 37.5 | 6.5        | 100.0 | 25.4   | 47.6 |
>
> Q3. **Data synthesis explanation.** The reviewer claims that "the self-play and tree search lack implementation details" and suggests that "using ReAct to generate thought-tool-input tuples" of solution data should be described in the method section.
>
> A3. Thanks for the suggestion. **Overall, the unsupervised data can be extracted by searching the solution data (tree search in ToolBench[3]) or merely prompting LLMs to trial (self-play in TALM[2]).** We initially don't describe the implementation of the self-play or tree search approach since we extract unsupervised data from the existing solution data synthesized by ToolBench[3]. Specifically, the solution paths are searched through a Depth First Search-based Decision Tree (DFSDT), which lets GPT-4 access different reasoning paths by choosing either to continue the current node or give up and expand a new node. For clarity, we have revised the paper to formalize the solution data (sequences of thoughts, tools, and inputs) in the method section (line 134) and describe the tree search approach for ToolBench data synthesis in the experiment section (line 357).
>
> Q4. **Relation with BERT.** The reviewer questions the analogy to BERT that "we are still training the LLMs under an autoregressive objective" and points out that "the model is trained on an augmented/generated dataset using BERT's 'masking' process but is not trained to predict what is missing in the context."
>
> A4. It's true that we are still training with next-token prediction autoregressive loss instead of predicting the masked token in the context. We mentioned the masked language models including BERT and Cloze[4] to elaborate the idea of predicting masked elements in the tool-use process (through meta-tasks). Such an idea shares a similar objective with predicting masked tokens in the context and enables the learning of the lurking knowledge beneath the unsupervised materials. We have clarified the idea above in our revised paper (line 146)
>
> Q5. **Other Concerns** about "the model version of ChatGPT", "full-parameter training", the Pass Rate metric based on ChatGPT, "directions of metrics", and the arrangement of tables and figures.
>
> A5. We appreciate the questions as well as the useful suggestions. (1) We adopt GPT-3.5-turbo-16k as ChatGPT throughout the evaluation. We evaluate the results of ChatGPT, GPT-4, and Claude-2 officially provided by ToolBench, but unfortunately, we can't find the versions of them in the ToolBench paper. (2) Sorry for the misdirection. We implement Lora training instead of full-parameter training. The "full-parameter" represents targeting all modules with Lora instead of just query and value matrixes. (3) The Pass-Rate metric is also evaluated with the help of ChatGPT to determine whether the query instructions are satisfied. Although the model can call the "Finish" tool to end the task and output a response, it does not necessarily satisfy the user's request. For example, it may respond "Sorry, I'm not able to retrieve the information." and should be evaluated as a failed task. (4) We have optimized our paper regarding the directions for metrics, typos, and paper layout. Please check our revised paper and kindly leave any further suggestions.
>
> [1] Schick, Timo, et al. "Toolformer: Language models can teach themselves to use tools." Advances in Neural Information Processing Systems 36 (2024).
>
> [2] Parisi, Aaron, Yao Zhao, and Noah Fiedel. "Talm: Tool augmented language models." arXiv preprint arXiv:2205.12255 (2022).
>
> [3] Qin, Yujia, et al. "Toolllm: Facilitating large language models to master 16000+ real-world apis." arXiv preprint arXiv:2307.16789 (2023).
>
> [4] Taylor, Wilson L. "“Cloze procedure”: A new tool for measuring readability." Journalism quarterly 30.4 (1953): 415-433.
>
> [5] Pearl, Judea. "Causal inference in statistics: An overview." (2009): 96-146.
>
> [6] Bareinboim, Elias, and Judea Pearl. "Meta-transportability of causal effects: A formal approach." Artificial Intelligence and Statistics. PMLR, 2013.

---

> ### Author Response · Authors · 2024-11-28
>
> Dear reviewer oF9y:
>
> I hope this message find you well. We have carefully considered your feedback and have made corresponding improvements to the manuscript. We truly value your insights, and your expertise has greatly contributed to enhancing the quality of our work. Could you please let us know if the revisions meet your expectations?  As the deadline for discussion nears, we kindly ask if you could review our rebuttal and updated paper. We are eager to address any additional questions that may arise. Thank you for your valuable support and consideration.
>
> Sincerely,
>
> Authors

---

### Official Review · Reviewer_Luu3 · 2024-11-05

**Soundness:** 3
**Presentation:** 3
**Contribution:** 2
**Rating:** 3
**Confidence:** 3

**Summary:**

This paper proposes to achieve generalizable tool learning by additionally training models on meta-reasoning QA tasks. The meta-reasoning data are constructed by asking questions about the tool-using process in multiple directions, including action effect, decision-making, reversion, action input boundary, etc. Experiment results show improved tool learning performance on tasks including SAW, BW, LOG, Toolbench and BFCL.

**Strengths:**

1. A novel meta-learning approach for tool learning showing improved tool-learning results.
2. The ablation study verifies the effectiveness of each meta-task.

**Weaknesses:**

1. In lines 224-226, "In order to maintain the general ability of the model in the first stage, only the parameters of the query and value projection layers of the Transformer are updated instead of full-parameter training." This constraint might also affect learning ability and make comparisons unfair. Results ensuring similar settings will make results more convincing.

2. The "LLaMA3-solution" baselines are updated fewer times (10k*3) compared with other models ((10k + 10k)*3). It would be fairer to test both training with more steps and using more solution data to ensure similar update steps.

3. Lack of 2-stage results on ToolBench and BFCL benchmarks, nor LLaMA3-8B-inst results on BFCL. Would be better to explain the setup more clearly.

4. Authors used the early stop to prevent overfitting, which also creates the possibility for larger variance due to the arbitrary selection of epoch numbers. Any more comprehensive results to eliminate this hyperparameter selection?

**Questions:**

See weaknesses

---

> ### Author Response · Authors · 2024-11-21
> **Rebuttal with additional experiment results and clarification**
>
> We appreciate the reviewer's constructive suggestions and concerns. We address the concerns in detail in the following lines.
>
> Q1.  **Ablations on hyper-settings.** The reviewer suggests that ensuring updating the same parts of model parameters (for baseline LLaMA3-2-stage) and the same "selection of epoch numbers" (we choose 1 epoch for MetaTool) will make results more convincing.
>
> A1. Thanks for the helpful suggestion to make our experiment more solid. **We configured those hyper-settings based on experimental results to achieve the best performance for each baseline method. Now we update their ablation results on ToolBench below in the table.** (1) Firstly, the results of both two variants of LLaMA3-2-stage (*-qv* targeting only query and value modules and *-full* targeting all parameter modules) are shown. The relatively weak performance of LLaMA3-2stage-full (-2.2%/-1.1% on average) suggests that training on metasets targeting full parameter modules may let the model overfit the QA tasks and hinder the subsequent training on solution data. We also observe some failed cases of LLaMA3-2stage-full that output meta-task answers instead of actions during testing. (2) Secondly, we show the results of LLaMA3-solution training with 1 epoch and MetaTool training with 2 epochs. The original LLaMA3-solution is trained with 2 epochs following the original configuration in ToolBench (ToolLLaMA). While early stopping for training merely on solution data harms the performance (-2.1%/-0.5% on average ), early stopping for MetaTool improves the performance (+5.6%/+4.7% on average). The contradiction is actually reasonable since the majority of the training data for MetaTool is QA data of meta-tasks (650k out of 776k). On the one hand, training too much on QA
>  data may cause overfitting (similar to the (1) case) and weaken the ability to plan actions. On the other hand, training on meta-tasks can bring sufficient knowledge about tools. That helps the LLMs to understand the expert solutions and learn the tool-use tasks faster, thus reducing the need for the second epoch training.
>
> **In summary, when the baseline settings above are configured to be the same as MetaTool, it shows a more significant advantage over LLaMA3-2stage-full (+17.4%/+9.4% on average) and LLaMA3-solution-1epoch (+10.8%/+7.7% on average).** The ablation results make our experiment more comprehensive, provide a fairer comparison, and verify that we have chosen the better hyper-settings for both LLaMA3-2-stage, LLaMA3-solution, and MetaTool. We have revised our paper to include those results in Table 4.
>
> |                        | I1-Inst. |      | I1-Tool |      | I1-Cat. |      | I2-Inst. |      | I2-Cat. |      | I3-Inst. |      | Averages |      |
> |---------------------:|---------:|-----:|--------:|-----:|--------:|-----:|---------:|-----:|--------:|-----:|---------:|-----:|---------:|-----:|
> | Models                 | Pass     | Win  | Pass    | Win  | Pass    | Win  | Pass     | Win  | Pass    | Win  | Pass     | Win  | Pass     | Win  |
> | LLaMA3-2stage-full     | 24.8     | 43.0 | 30.0    | 43.9 | 36.0    | 43.0 | 29.2     | 52.1 | 28.9    | 37.7 | 23.7     | 56.8 | 28.5     | 46.1 |
> | LLaMA3-2stage-qv       | 31.4     | 43.6 |  35.6   | 44.8 | 40.3    | 44.0 | 40.4     | 48.0 | 36.1    | 46.8 | 28.5     | 58.0 | 34.7     | 47.2 |
> | LLaMA3-solution-1epoch | 30.9     | 45.0 | 37.3    | 44.9 | 34.1    | 42.0 | 39.5     | 51.3 | 36.0    | 42.4 | 32.8     | 61.0 | 35.1     | 47.8 |
> | LLaMA3-solution (2epochs)    | 32.1     | 45.3 | 39.0    | 43.9 | 36.4    | 43.0 | 40.1     | 52.5 | 40.1    | 43.4 | 35.6     | 61.8 | 37.2    | 48.3 |
> | MetaTool-2epoch       | 35.7     | 44.2 | 35.6    | 43.7 | 39.0    | 47.6 | 45.6     | 51.5 | 46.1    | 49.5 | 39.5     | 68.3 | 40.3     | 50.8 |
> | MetaTool (1epoch)              | 42.5     | 52.1 | 41.8    | 51.3 | 43.3    | 46.1 | 52.0     | 54.9 | 50.0    | 54.0 | 45.5     | 74.5 | 45.9     | 55.5 |
>
> Q2. **Training with different update steps.** The reviewer poses the concern that "the LLaMA3-solution baselines are updated fewer times (10k*3) compared with other models ((10k+10k)*3)" and suggests that "It would be fairer to train the baseline with more steps and more solution data to ensure similar update steps."
>
> A2. **We insist that the comparison with this configuration is fair, since the main contribution of our method is exactly generating additional high-quality data for training without any human annotation.** Both LLaMA3-solution and MetaTool are trained with supervised fine-tuning with 3 epochs, thus more training data naturally leads to more update steps. The superior performance of MetaTool verifies the effectiveness of the controlled variable (i.e. additional meta-task data) as well as our contribution. Also, it would be less fair if we train LLaMA3-solution with more solution data, since then the comparison will be 10k meta-task data against 10k solution data.

---

> > ### Comment · Reviewer_Luu3 · 2024-11-27
> > **Reviewer response**
> >
> > A1: It is weird that "LLaMA3-solution" generally increases with more epochs, but "MetaTool" significantly decreases with more epochs. ($\sim$ 5 points on average). So it seems epoch number is an essential hyperparameter affecting the performance gain (much larger gap on 2 epochs than 1 epoch). Also the qv version of llama3-solution and metatool are not shown for fairer comparison.
> >
> > A2: the update steps plays a significant role in model performance so the argument is still not convincing. Or the argument should be explained clearer in the draft.

---

> > > ### Author Response · Authors · 2024-11-29
> > >
> > > Thank you for your response. We greatly appreciate your concerns and suggestions to make our experiment more comprehensive. There may still be some misunderstandings, and we would like to further clarify them.
> > >
> > > Q1. **Weird performance gain with different epoch numbers.**
> > >
> > > A1. As we have analyzed in our rebuttal comment, **MetaTool's performance decreases with more epochs due to overfitting on the meta-tasks. It is important to understand that there is a domain shift between meta-tasks and the original tool-use tasks.** While meta-tasks ask questions about tool understanding (examples shown in Figure 4), tool-use tasks require the model to take tool-calling actions to achieve goals (showcased in Figure 6). Given that the majority of the training data consists of meta-tasks, training on it for multiple epochs weakens the model's ability to plan actions. From the perspective of meta-learning [1], meta-tasks are designed to help LLMs learn the original tool-use task better. **Thus it's natural to adjust the training epochs to avoid overfitting on individual tasks[2,3] (i.e. meta-tasks in our case).**
> > >
> > > We indeed observe that the epoch number considerably affects MetaTool on ToolBench. **The rationale behind that is data imbalance, with meta-tasks comprising roughly 60% tokens of the total data.** With less meta-tasks data or more solution data, MetaTool can be more robust to training epochs. However, less meta-tasks data can also lead to less improvement in the first epoch training. It's worth exploring the best configuration to achieve the best performance for different tasks and toolsets. Also, it is beneficial that MetaTool does not require more epochs on the solution data and benefits the generalization (as suggested by results on BFCL), as the meta-tasks help it understand and learn the solution paths better.
> > >
> > > Q2. **Lack of qv version of LLaMA3-solution and MetaTool.**
> > >
> > > A2. There may be a misunderstanding of our method. **Targeting the qv modules is a design tailored for our 2-stage model and cannot be applied to 1-stage models like MetaTool in practice.** The idea is to target the qv modules in the first stage (trained on meta-tasks data) and target all modules in the second stage. However, when training MetaTool, the meta-tasks and solution data are mixed in every batch for model updates, making it impossible to target different parameter modules during loss calculation. For clarity, we will revise our paper to explain our method design more clearly.
> > >
> > > Q3. **Discussion about the update steps (iterations).** The reviewer argues that it is not convincing to train the LLaMA3-solution baseline on 10k solution data ($P$) for 3 epochs while training MetaTool on 10k solution data and 10k meta-tasks data ($P+M$) for 3 epochs as a comparison (in the tool-oriented scenario).
> > >
> > > A3. We would like to clarify that **we implemented the LLaMA3-solution baseline and compared it with MetaTool as an ablation study to answer the question: "Does augmenting the solution data with additional meta-tasks data improve the tool-use ability of LLMs?"** The results answer this question and verify the effectiveness of our self-supervised data augmentation method compared to other training paradigms shown in Figure 1. Additionally, this ablation setting is commonly used in data augmentation research, such as in TinyBERT[4], MAE[5], and EDA[6]. For different data augmentation methods, the dataset composition is also part of the methodology. Ablating the data composition and training with SFT for the same epoch number ensures a fair comparison. It is in reinforcement learning research, not ours, where maintaining the same update steps is typically used to ensure fair settings.
> > >
> > > [1] Finn, Chelsea, Pieter Abbeel, and Sergey Levine. "Model-agnostic meta-learning for fast adaptation of deep networks." International conference on machine learning. PMLR, 2017.
> > >
> > > [2] Yin, Mingzhang, et al. "Meta-learning without memorization." arXiv preprint arXiv:1912.03820 (2019).
> > >
> > > [3] Guiroy, Simon, et al. "Improving meta-learning generalization with activation-based early-stopping." Conference on lifelong learning agents. PMLR, 2022.
> > >
> > > [4] Jiao, Xiaoqi, et al. "Tinybert: Distilling bert for natural language understanding." arXiv preprint arXiv:1909.10351 (2019).
> > >
> > > [5] He, Kaiming, et al. "Masked autoencoders are scalable vision learners." Proceedings of the IEEE/CVF conference on computer vision and pattern recognition. 2022.
> > >
> > > [6] Wei, Jason, and Kai Zou. "Eda: Easy data augmentation techniques for boosting performance on text classification tasks." arXiv preprint arXiv:1901.11196 (2019).

---

> > > ### Author Response · Authors · 2024-11-30
> > >
> > > I hope this message finds you well. We have carefully considered your feedback and have provided corresponding explanations and references. We truly value your insights, and your expertise has greatly contributed to enhancing the quality of our work. Could you please let us know if the response addresses your concern? We are eager to address any additional questions that may arise. Thank you for your valuable support and consideration.

---

> ### Author Response · Authors · 2024-11-21
>
> Q3. **Lack of 2-stage results.** The reviewer points out the "lack of 2-stage results on ToolBench and BFCL benchmarks, nor LLaMA3-8B-instuct results on BFCL".
>
> A3. Thanks for pointing that out. **We now update the results of LLaMA3-2-stage and LLaMA3-8B-instruct on BFCL below in the table and have updated them in our paper in Table 5.** LLaMA3-2-stage here is trained on ToolBench first on the 650k meta-task data with 1 epoch then on the 126k solution data with 1 epoch, in order to have the same update steps with MetaTool. We adopt the results of LLaMA3-8B-instruct officially released by BFCL recently and test the LLaMA3-2-stage ourselves. The results show that the zero-shot generalizability of MetaTool is improved in most testing sets compared with its base model LLaMA3-8B-instruct (+4.3% success rate on average). We notice that MetaTool also shows retrogression in testing sets such as *multi* and *live-parallel*. That is attributed to merely training in the ToolBench scenario which is similar to the *simple* testing set (detailed explained in our paper lines 421-427). Training in 2 stages partly reduces the zero-shot ability of the model.
>
> |                    | nonlive-AST |          |          |      | live-AST |          |          |      | Multi-turn | Hal.  |        | Ave. |
> |:------------------:|:-----------:|:--------:|:--------:|:----:|:--------:|:--------:|:--------:|:----:|:----------:|:-----:|:------:|:----:|
> |                    | simple      | multiple | parallel | M&P  | simple   | multiple | parallel | M&P  | base       | rel.  | irrel. |      |
> | LLaMA3-8B-instruct | 63.1        | 85.5     | 51.5     | 44   | 60.9     | 60.8     | 37.5     | 20.8 | 3          | 75.6  | 27.4   | 42.3 |
> | LLaMA3-2-stage     | 66.8        | 60.0     | 5.0      | 6.0  | 53.9     | 33.1     | 16.8     | 6.3  | 5.0        | 98.1  | 10.5   | 41.9 |
> | MetaTool           | 78.3        | 55.0     | 66.0     | 63.5 | 58.1     | 50.1     | 18.8     | 37.5 | 6.5        | 100.0 | 25.4   | 47.6 |

---

> ### Comment · Reviewer_Luu3 · 2024-11-27
> **Response**
>
> Thanks, this makes results more comprehensive. Similar results should be shown on Toolbench.
>
> I increased soundness score.

---

> > ### Author Response · Authors · 2024-11-27
> >
> > Thanks for your positive response. We have included the results of these models (e.g. LLaMA3-8B-instruct, LLaMA3-2-stage) in Table 3. Please check them out in our paper.

---

### Meta-Review · Area_Chair_RzpF · 2024-12-16

**Metareview:**

This paper introduces MetaTool, a novel approach to enhance large language models’ (LLMs) ability to use tools. Unlike traditional methods that rely on prompts or labeled data, MetaTool employs self-supervised learning through six meta-tasks: Effect, Decision-making, Reversion, Input Boundary, Output Boundary, and Counterfact. These tasks teach foundational concepts like cause and effect, permissible actions, and expected outcomes. MetaTool demonstrates strong performance across tool-based tasks, rivaling models like ChatGPT in planning and chat scenarios.

While empirical results demonstrate the effectiveness of the method, most experiences have been done in simulated environments, leaving questions on the applicability of the proposed approach in more realistic scenarios. In addition, reviewers noted some ambiguity in the methodological details and the clarity on fair comparisons.

**Additional Comments On Reviewer Discussion:**

The authors explained and added more experiments to address the concerns raised by the reviewers. One reviewer raised the score from 5 to 6

---

### Decision · Program_Chairs · 2025-01-22

Reject